# Applying Metagenomic Analysis Using Nanopore Sequencer (MinION) for Precision Medicine in Bacterial Keratoconjunctivitis: Comprehensive Validation of Molecular Biological and Conventional Examinations

**DOI:** 10.3390/ijms24032611

**Published:** 2023-01-30

**Authors:** Hiroshi Eguchi, Fumika Hotta, Shunji Kusaka

**Affiliations:** Department of Ophthalmology, Kindai University, Osakasayama 589-8511, Japan

**Keywords:** bacterial keratoconjunctivitis, corneal scraping, culture, smear microscopic examination, nanopore sequencer (MinION)

## Abstract

Smear microscopic examination and culture of the corneal scrapings are the gold standards for the diagnosis of bacterial keratoconjunctivitis. High-sensitivity molecular biological examinations of the ocular surface specimens are used clinically. However, the results require careful interpretation to avoid the unintentional detection of indigenous bacteria. Results of conventional and state-of-the-art examinations require clinical verification for specificity and sensitivity. In this study, smear microscopic examination, culture, and nanopore sequencing using the MinION of ocular surface specimens from eight clinically diagnosed bacterial keratoconjunctivitis cases were performed and compared. Seven of the eight cases (87.5%) were smear positive and five (62.5%) were culture positive. The former showed the same genus in >60% of the classified reads as one specific bacterium inferred from the smear microscopy when sequenced by the MinION. In two of the three culture-negative cases, the smear-positive images were highly reminiscent of the species comprising most of the MinION sequences. Four of the five culture-positive cases were consistent with the most prevalent bacteria in the sequencing results. Probable contamination among specimens processed on the same day were observed. In conclusion, the microscopic examination of the corneal scraping specimens may be more sensitive and specific than the culture examination. Additionally, although metagenomic analysis using the MinION contributes to more precise medication for bacterial keratoconjunctivitis, contamination can affect the results.

## 1. Introduction

Bacterial keratitis is a severe ocular disease that can cause blindness [1,2]. The causative bacterial species vary and the findings of slit-lamp microscopy, clinical course, drug susceptibility, and visual prognosis differ depending on the causative agent. Bacterial conjunctivitis is a common infection [3,4]. It does not directly cause blindness. Most cases are relatively mild; however, some can result in keratitis due to an immune response. Early identification of the causative bacterium and its treatment leads to early resolution of the inflammation resulting in a favorable visual prognosis. Smear microscopy and culture of the ocular surface specimens are considered the gold standard for identifying the pathogenic strain of bacterial keratoconjunctivitis [5]. Although the former is the most rapid test, its sensitivity and specificity depend on the examiner’s level of expertise. Using smear microscopy, only Gram-positive or -negative, cocci or rods can be identified. Therefore, this test is mostly limited to identification at the genus level without the identification of drug susceptibility. As such, smear microscopy tests are only a tentative basis for selecting antimicrobial agents.

The culture of the ocular surface specimens isolates viable bacteria from the specimen, indicating their survival and presence in the specimen. When a specific bacterium is isolated from the corneal scraping or the eye discharge, a specific diagnosis of bacterial keratoconjunctivitis can be made. It is often possible to identify the species of bacteria using the culture of ocular surface specimens. In most cases, drug susceptibility can also be determined. This can provide an objective and accurate basis for selection of antimicrobial agents. However, the culture of ocular surface specimens can produce negative results even with the appropriate culture conditions and the collection of large amounts of material from the specimen because optimal conditions for the culture differ depending on the bacterial species. Although they are primarily environmental bacteria, there are many difficult-to-culture or viable but non-cultivable bacteria [6]. There is no guarantee that these bacteria will not become pathogens on the ocular surface when exposed to the outside. A combination of the smear microscopy and the culture methods can be applied for rigorous diagnosis of bacterial keratoconjunctivitis.

Molecular biological methods such as polymerase chain reaction (PCR) have been introduced for the clinical diagnosis of ocular infections by detecting the genetic information of a small number of the microorganisms in a specimen [7,8,9,10]. The greatest advantage of such methods is their high detection sensitivity, as they amplify even minute amounts of the DNA in the specimens such as corneal scrapings. However, there is always the risk of the simultaneous detection of specimen contamination. The PCR targeting a specific bacterium can only reveal the presence or the absence of that bacterium. On the ocular surface exposed to the outside world, it is assumed that there is always external contamination. It is unclear whether the PCR targeting specific well-known bacteria can detect true pathogenic strains.

In recent years, the metagenomic analysis of clinical specimens using next-generation sequencing (NGS) technology has been introduced as a novel tool for identifying the causative bacteria by verifying the relative abundance of DNA in a large population of bacteria, leading to a diagnosis [11]. Although it is still far from being clinically used in routine examinations [12], a particular bacterium is assumed to be the causative agent when its relative abundance in an acute infection specimen is high. However, if the DNA read counts of multiple bacteria in a chronic infection are almost equal, the results must be interpreted with caution and comprehensively in conjunction with the conventional smear and the culture results. Most of the NGS equipment is not portable enough and is costly and time consuming for clinical use.

In 2014, Oxford Nanopore Technologies launched a palm-sized sequencer, called MinION, which was capable of performing metagenomic analysis at a low cost and rapid rate. The results achieved using the MinION were equivalent to those of the world-acclaimed next-generation sequencer, Illumina Hiseq [13]. Since then, the MinION has been introduced into clinical practice [14,15,16,17], however to the best of our knowledge, no comprehensive interpretation of the results of the smear microscopy, the culture, and metagenomic analysis of the same ocular infectious specimens has been reported. In this study, we collected the ocular surface specimens from eight patients with clinically diagnosed bacterial keratoconjunctivitis and subjected them to the smear microscopy, the culture, and metagenomic analysis using the MinION to verify their sensitivity and specificity.

The purpose of this study was to provide evidence for how the conventional examinations and the state-of-the-art molecular biological tests should be interpreted for precision medicine in bacterial keratoconjunctivitis.

## 2. Results (Table 1)

### 2.1. Smear Microscopy

Seven of the eight cases (87.5%) included in this study were smear positive (Figure 1a, Figure 2a, Figure 3a, Figure 4a, Figure 5a, Figure 6a and Figure 8a). Gram-negative rods were detected in three cases, two of which were suspected to be Moraxella lacunata (Figure 1b and Figure 2b). The remaining case was of elongated Gram-negative rods, reminiscent of the Pseudomonas species (Figure 3b). Cocci were detected in three cases, two of which were Gram-positive and one of which was Gram-negative. The two cases of Gram-positive cocci were reminiscent of the genus Streptococcus, due to the identification of the chain cocci and the somewhat heterogeneous Gram staining (Figure 4b and Figure 5b). One of these was an encapsulated diplococcus, reminiscent of Streptococcus pneumoniae (Figure 5b). Gram-negative diplococci were detected in one specimen from a case which was clinically highly likely to be gonococcal keratoconjunctivitis (Figure 8b).

**Table 1 ijms-24-02611-t001:** Case details.

Case	Age	Sex	Smear	Culture	Dominant Species ^6^	Reads ^7^	Ave. ^8^	Abundance (%) ^9^
1	45	M	GNR ^1^	Negative	*M. lacunata*	832	2000	70.5
2	73	F	GNR	Negative	*M. lacunata*	24,969	2000	72.2
3	76	M	GNR	*P. aeruginosa*	*P. aeruginosa*	2357	2000	95.4
4	56	M	GPC ^2^	SDSE ^5^	SDSE	982	2000	94.0
5	83	M	GPC	*Str. pneumoniae*	*Str. pneumoniae*	25	1988	64.0
6	79	F	GPR ^3^	*Corynebacterium*	None	93	1187	44.1
7	47	M	Negative	Negative	*S. aureus*	61	2000	81.9
8	38	M	GNC ^4^	*N. gonorrhoeae*	*N. gonorrhoeae*	5335	2000	97.8

^1^ GNR: Gram-negative rod, ^2^ GPC: Gram positive cocci, ^3^ GPR: Gram positive rod, ^4^ GNC: Gram negative cocci, ^5^ SDSE: *Streptococcus dysgalactoae* subsp. *equisimilis*, ^6^ the strains with the number of reads represented as more than 50% by relative abundance on the MinION, ^7^ classified reads, ^8^ average sequence length, and ^9^ abundance ratio of the most dominant species.

**Figure 1 ijms-24-02611-f001:**
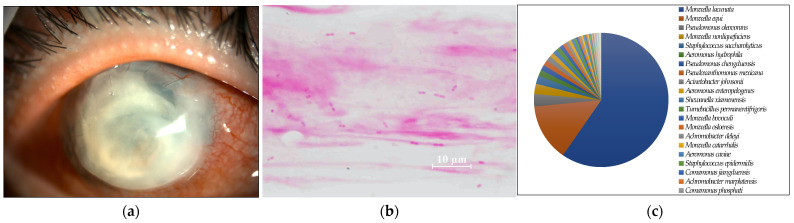
(**a**) Anterior segment photograph. The entire graft is opaque due to infection after corneal transplantation. (**b**) Smear microscopy. Many large Gram-negative rods were detected. (**c**) Nanopore sequencing results (relative abundance) of Case 1. *Moraxella lacunata* was detected as the dominant species by MinION.

**Figure 2 ijms-24-02611-f002:**
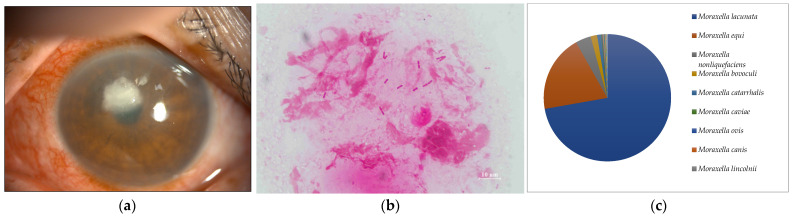
(**a**) Anterior segment photograph. A paracentral corneal abscess is present. (**b**) Smear microscopy. Several large Gram-negative rods were detected. (**c**) Nanopore sequencing results (relative abundance) of Case 2. *Moraxella lacunata* was detected as the dominant species by MinION.

**Figure 3 ijms-24-02611-f003:**
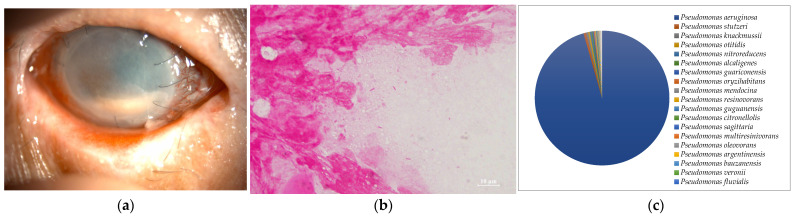
(**a**) Anterior segment photograph. The entire cornea has pale opacity and hypopyon is obvious. (**b**) Smear microscopy. Elongated Gram-negative rods were detected. (**c**) Nanopore sequencing results (relative abundance) of Case 3. (**c**) An overwhelming majority of *Pseudomonas aeruginosa* was detected by MinION.

**Figure 4 ijms-24-02611-f004:**
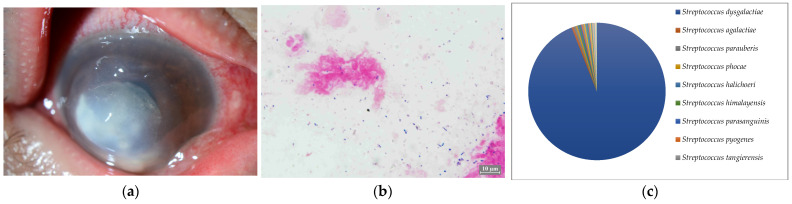
(**a**) Anterior segment photograph. The lower 3/4 of the flap created by former LASIK forms a circular abscess. (**b**) Smear microscopy. Gram-positive chain cocci were detected. (**c**) Nanopore sequencing results (relative abundance) of Case 4. An overwhelming majority of *Streptococcus dysgalactoae* was detected by MinION.

**Figure 5 ijms-24-02611-f005:**
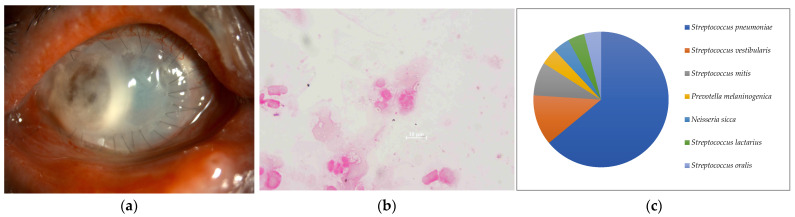
(**a**) Anterior segment photograph. The entire corneal graft was opaque, with abscess and infiltration, and temporal corneal graft showed melting. (**b**) Smear microscopy. Diplococci that appear Gram-negative were determined primarily due to heterogeneous staining but were found to be encapsulated Gram-positive. (**c**) Nanopore sequencing results (relative abundance) of Case 5. An overwhelming majority of *Streptococcus pneumoniae* was detected by MinION.

**Figure 6 ijms-24-02611-f006:**
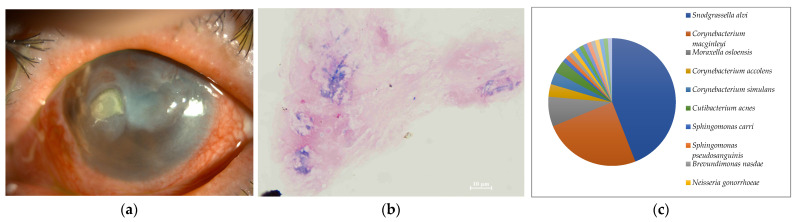
(**a**) Anterior segment photograph. Corneal infiltration and abscess were found in nasal cornea, suggestive of infectious keratitis. (**b**) Smear microscopy. Few Gram-positive rods are detected (**c**) Nanopore sequencing results (relative abundance) of Case 6. No bacterial species with a relative abundance greater than 50% was detected by MinION.

**Figure 7 ijms-24-02611-f007:**
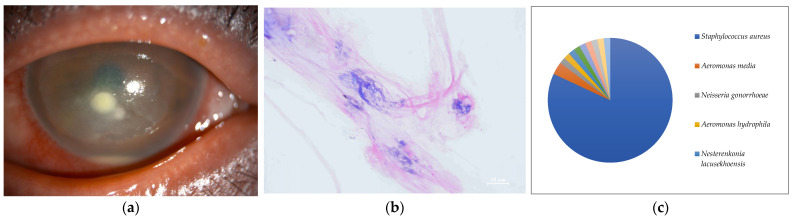
(**a**) Anterior segment photograph. Corneal infiltration and corneal abscess were found, suggestive of infectious keratitis. (**b**) Smear microscopy. No organisms were detected by Gram staining. (**c**) Nanopore sequencing results (relative abundance) of Case 7. An overwhelming majority of *Staphylococcus aureus* was detected by MinION.

**Figure 8 ijms-24-02611-f008:**
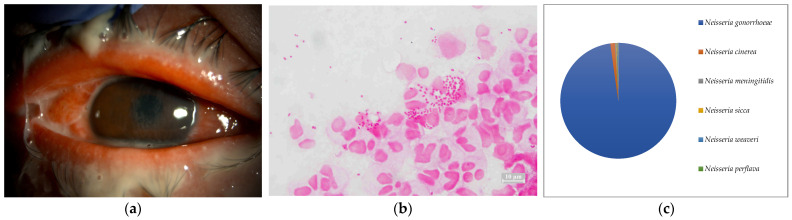
(**a**) Anterior segment photograph. Severe muco-purulent eye discharge and conjunctival hyperemia were found. The immune response caused corneal infiltration. These findings are mostly suggestive of gonococcal keratoconjunctivitis. (**b**) Smear microscopy. Gram staining of ocular surface aspirates containing eye discharge detected abundant Gram-negative diplococci. (**c**) Nanopore sequencing results (relative abundance) of Case 8. An overwhelming majority of *Neisseria gonorrhoeae* was detected by MinION.

### 2.2. Culture

Five of the eight cases (62.5%) included in this study were culture-positive. *Streptococcus dysgalactoae* subsp. *equisimilis* (SDSE) and *Streptococcus pneumonia* were isolated from two cases in which Gram-positive chain cocci were detected using the smear microscopy. *Pseudomonas aeruginosa* was isolated from a patient with elongated Gram-negative rods. *Corynebacterium* and *Neisseria gonorrhoeae* were each isolated in single cases in which only a few Gram-positive rods and several Gram-negative diplococci were detected, respectively.

### 2.3. Nanopore Sequencing by the MinION

In seven of the eight cases (Cases 1–8) (87.5%) included in this study, one dominant strain with a relative abundance of 60% or more was detected. The average abundance of the dominant species was 77.5%, which ranged from 44.1% to 97.8%. The sequence length of the eight specimens was 1187 to 2000 base pair (bp), with an average of 1897 bp, which is standard for the bacterial 16S rRNA gene. Although the culture was negative in Cases 1 and 2, where Gram-negative rods were detected using the smear microscopy, *M. lacunata*, which is compatible with the smear microscopic image was detected as the dominant species by the MinION (Figure 1c and Figure 2c). In Cases 3, 4, 5, and 8, the results of the smear microscopic images, the culture, and the nanopore sequencing were compatible (Figure 3c, Figure 4c, Figure 5c and Figure 8c). In Case 6, where the nanopore sequencing did not detect any species with a relative abundance greater than 60%, *Snodgrassella alvi* was the most dominant species (44.1%), followed by *Corynebacterium macginleyi* (24.7%) and *Moraxella osloensis* (7.5%). In Case 7, where both the smear microscopy and the culture were negative, the nanopore sequencing showed *Staphylococcus aureus* to be the dominant species (Figure 7c).

## 3. Discussion

To the best of our knowledge, this is the first study that performed a metagenomic analysis of the ophthalmic clinical specimens using the portable nanopore sequencer (the MinION), presented the smear microscopic results of all cases, and compared the cultures and the clinical information in detail. The utility of new molecular biological techniques and most conventional smear microscopes has been demonstrated. Microscopic images of the smear can identify the genus level with high probability based on Gram staining, morphology, and the epidemiological background of the case. Species-level identification is possible in some cases. For example, large Gram-negative rods were detected in Cases 1 and 2, which were culture-negative. Empiric therapy targeting *Moraxella* could be initiated even if the culture was negative because of its characteristic smear image. Cases 1 and 2 were treated with only the levofloxacin (LVFX) ophthalmic drops, resulting in the rapid disappearance of inflammation because the clinical ophthalmic isolates of *Moraxella* in Japan have good quinolone sensitivity [18]. Definitive treatment could be initiated without the culture results, which highlights the usefulness of the smear microscopy. Bergey’s Manual states that the addition of plasma or body fluid is necessary to culture *Moraxella*, the culture temperature should be slightly lower than that for other bacteria, and microaerobic conditions, as for *Haemophilus*, are preferable [19]. It should be noted that the cultures of the corneal scrapings could be negative for *Moraxella lacunata* keratitis.

All of the most dominant organisms in the smear specimens, the culture, and the metagenomic analysis matched in Cases 3, 4, 5, and 8. In all cases, it was not difficult to identify the pathogen from the smear images and the patient’s past history, and the empirical treatment was initiated promptly, which was also confirmed by the culture results. Thus, the clinical metagenomic analysis was not necessary in these cases. However, the culture usually requires 24 h. Further testing after isolation often takes several days. Therefore, species identification is not clinically possible until the next day at the earliest, or several days after the specimen submission in some cases. In this study, we set the sequencing run time of the MinION to 12 h. In Cases 3, 4, 5, and 8, the average number of detected most dominant bacterial reads was 87.8% (64–97.8%). Therefore, even with a shorter sequence runtime (e.g., 30 min or 1 h), it is highly likely that the same species of bacteria were detected as the most dominant bacteria. A shorter sequence runtime may be useful in acute infections for the clinical introduction because it may provide the species identification results much earlier than the culture results [20].

The usefulness of the metagenomic analysis was highlighted in Case 7, wherein both the smear specimens and the cultures were negative. The patient presented with a history of atopic dermatitis and laser in situ keratomileusis (LASIK) in his eye. He was treated for the corneal epithelial defect around the edge of the corneal flap created during the previous LASIK, which was treated as a geographic corneal ulcer suggestive of herpetic keratitis. Subsequently, a corneal abscess appeared after the steroid eye drops were added. Based on the findings of the slit-lamp microscopy and a systemic history of atopic dermatitis, which has a high probability of harboring *Staphylococcus aureus* [21], empirical medication was initiated with frequent chloramphenicol eye drops, targeting the corneal abscess. Simultaneously, the partially melted cornea near the flap was scraped for the smear and the culture. However, both examinations yielded negative results. Thereafter, the corneal ulcer gradually disappeared, and the keratitis was successfully eliminated. Thus, the infection was almost certainly caused by chloramphenicol-sensitive bacteria, in particular, Gram-positive bacteria. The metagenomic analysis of this case showed that *S. aureus* was the most dominant bacterial species, with a relative abundance of 81.9%. The patient’s history, and the clinical findings and the course were consistent with the presence of *S. aureus* infection. Qualitative PCR testing of a specific bacterium only provides information on its presence; if a bacterium is found to have a large number of genes in multiplex quantitative PCR, it is likely to be the pathogen; however, information on only a few selected microorganisms is available. Case 7 was most likely caused by the well-known bacterium, *S. aureus*, but the metagenomic analysis had the advantage of being able to detect unexpected bacteria as pathogens because of its comprehensive analysis.

Case 7 also produced results highlighting the contamination, which is a weakness of the metagenomic analysis. Similarly, the results of Case 6 appeared to be contaminated. In Case 6, the examiner (HE) had scraped the cornea of another patient with Acanthamoeba keratitis on the same day, in the same time zone and at the same location (ophthalmology outpatient treatment room). The latter patient had similar specimens collected on that day and on other three days, and each time *Snodgrassella alvi* was detected as the predominant species in the corneal scraping specimens. *S. alvi* is a honeybee gut commensal [22], and there are no known reports of its isolation from the ocular surface of humans. The patient was a long-time regular user of cosmetics containing natural ingredients of unknown details which may contain substances related to bee venom. The corneal scraping specimen from Case 6 was probably contaminated due to the specimen from this patient during the corneal scraping procedure. The specimens from Cases 6 and 7 contained only one read (relative abundance of 1.1% and 1.6%, respectively) of *Neisseria gonorrhoeae* DNA. Although the sample collection dates for Cases 6, 7, and 8 were completely different, the same examiner (HE) performed the DNA extraction from the specimens and library preparation for the MinION for 10 days (on the same day for Cases 7 and 8). It is presumed that specimens from Cases 6 and 7 were contaminated with the gonococcal DNA of the Case 8 sample during the experimental manipulation. It should be noted that contamination of such specimens can affect the results in the metagenomic analysis of ophthalmic clinical specimens, which were originally low in DNA content.

The limitations of this study were the small sample size and the sensitivity of the smear specimen which is affected by the examiner’s expertise. In further studies, increasing the sample size and applying the same techniques used in this study to not just acute but also chronic infections will elucidate the usefulness and the factors to be considered in metagenomic analyses for ocular infections.

## 4. Materials and Methods

Of the infectious keratoconjunctivitis patients treated by the author (HE) from October 2021 to October 2022, 8 patients consented to this study and became candidates. After administration of eye drop anaesthesia (oxybuprocaine hydrochloride ophthalmic solution 0.4%, Benoxil^®^ ophthalmic solution, Santen Pharmaceutical Co., Ltd., Osaka, Japan), the cornea was collected by scraping with a sterile spatula knife. The eye discharge was collected using a sterile cotton swab. The scraped cornea was smeared on a glass slide (Super Frost Glass, Matsunami Glass Ind., Ltd., Osaka, Japan), stained with Gram stain reagents (Favor G “Nissui,” Nissui Pharmaceutical Co., Ltd., Tokyo, Japan), and observed under an optical microscope (Nikon ECLIPSE 80*i*, Nikon Corporation, Tokyo, Japan) at 1000× magnification with immersion oil. Another sample set of scraped corneal tissue and ocular surface aspirate was collected by pumping 200 μL of PBS (Gibco™, PBS, Thermo Fisher Scientific K.K., Tokyo, Japan) with a pipettor, washing the ocular surface, and aspirating. The collected samples were dispensed into two sterilised sample tubes (Eppendorf Safe-Lock tube, Biopur^®^, Eppendorf SE, Hamburg, Germany), one of which was sent to the bacteriological laboratory at Kindai University Hospital on the same day for culture. Bacterial species identification was performed using MALDI-TOF MS after isolating the bacteria from the specimen. Another collected specimen was stored at −80 °C until metagenomic analysis.

For metagenomic analysis, DNA was extracted from stored samples thawed at room temperature using the AllPrep PowerViral DNA/RNA Kit (^©^QIAGEN, Hilden, Germany) according to the manufacturer’s instructions. The DNA concentration was measured using a Qubit 4 fluorometer (Invitrogen™, Thermo Fisher Scientific). Total 10ng genomic DNA was adjusted to 10 μL by volume using Nuclease-free water. The 16Sregion was amplified according to the manufacturer’s instructions using a 16S Barcoding Kit (SQK-RAB204, Oxford Nanopore Technology, Oxford, UK). Amplification was performed in 50 μL reactions containing 10 μL of genomic DNA, 25 μL of LongAmp Taq 2X master mix (M0287, NEB), 1 μL of 16S Barcode, and 14 μL of Nuclease-free water. Thermal cycling comprised an initial denaturation step (1 min at 95 °C), followed by 25 cycles of denaturation (20 s at 95 °C), annealing (30 s at 55 °C), and 2 min extension at 65 °C. The final extension was performed for 5 min at 65 °C. DNA purification was performed using AMPure XP beads (Beckman Coulter, Inc., Brea, CA, USA). A sequence library was created according to the manufacturer’s instructions. The library was loaded into a flow cell R9.4.1 (FLO-MIN106D, Oxford Nanopore Technology, UK) and sequenced with MinION Mk1C (Oxford Nanopore Technologies, UK) for 12 h. The obtained Fastq data were analysed using Fastq 16S workflow v2022.01.07 of EPI2ME (Oxford Nanopore Technologies, UK) with default settings. A pie chart of identified bacterial species was created using Excel.

## 5. Conclusions

Metagenomic analysis using a nanopore sequencer (MinION) is useful for diagnosing ocular surface infections. However, ocular surface infections should not be diagnosed solely based on the results of the nanopore sequencing because contamination can affect the results. Ophthalmologists should have a precise understanding of the sequencing conditions and the analysis results, and accumulate the verification results of the cases comprehensively examined using the smear microscopy and the culture methods.

## Data Availability

The data presented in this study are available upon request from the corresponding author. The data were not publicly available because of privacy concerns.

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
