# Peer review of "Applying Metagenomic Analysis Using Nanopore Sequencer (MinION) for Precision Medicine in Bacterial Keratoconjunctivitis: Comprehensive Validation of Molecular Biological and Conventional Examinations"

_ijms, 2023, doi:10.3390/ijms24032611_

Round 1
Reviewer 1 Report
This study compared the smear microscopic examination, culture, and nanopore sequencing using MinION of ocular surface specimens from eight clinically diagnosed bacterial keratoconjunctivitis cases. They found that the traditional microscopic examination of corneal scraping specimens is more sensitive and specific than the culture examination. Although metagenomic analysis may be more precise potential contamination could occur.
Overall, the results obtained are not new. A short report would be more appropriate. Besides, the material and method section can be improved to include more details about experiment design, sample processing, data analysis, statistics, etc.
Author Response
1) Comments and Suggestions for Authors
This study compared the smear microscopic examination, culture, and nanopore sequencing using MinION of ocular surface specimens from eight clinically diagnosed bacterial keratoconjunctivitis cases. They found that the traditional microscopic examination of corneal scraping specimens is more sensitive and specific than the culture examination. Although metagenomic analysis may be more precise potential contamination could occur.
Overall, the results obtained are not new. A short report would be more appropriate. Besides, the material and method section can be improved to include more details about experiment design, sample processing, data analysis, statistics, etc
Response:
As you pointed out, metagenomic analysis using NGS technology is much more sensitive than culture in diagnosing infectious diseases, and that contamination during manipulation affects the results in samples with low microbial loads. However, as far as we know, there are no reports that have performed metagenomic analysis of ophthalmic clinical specimens using the portable nanopore sequencer, MinION, presented the all smear microscopic results of all cases, and compared cultures and clinical information in detail.
Articles on new diagnostic techniques tend to emphasize their advantages; however this manuscript presents the results of all classical tests in all cases and compares the advantages and disadvantages of the new diagnostic techniques in detail. For example, in Case 6, MinION detected no organisms with relative abundance greater than 50% and a small number of gram-positive rods on both smear microscopy and culture, could have been diagnosed as the world's first case of Snodgrassella alvi keratitis if the sequencing results were taken at face value. However, based on interviews, literature searches, and the clinical course of the case, we ruled it out. Similarly, in Case 7, a trace amount of gonococcal DNA was detected in the specimen; however, clinically gonococcal infection could be completely ruled out, and we have disclosed the negative aspect that the specimen was processed on the same day as Case 8, in which gonococcal infection was evident, which may have contaminated the sample. We believe that this paper is novel in that it warns against the dangers of highly sensitive molecular biological tests with both detailed case presentations, including all smear specimens, and disclosure of specific sample manipulation processes.
We have revised the first sentence of “Discussion” section. The new sentence is as follows; “To the best of our knowledge, this is the first study that performed metagenomic analysis of ophthalmic clinical specimens using the portable nanopore sequencer, MinION, presented smear microscopic results of all cases, and compared cultures and clinical information in detail.”
Although we intended to describe the specific reproducible methods, we have added the sentences “Amplification was performed in 50 μl reactions containing 10 μl of genomic DNA, 25μl of LongAmp Taq 2X master mix (M0287, NEB), 1 μl of 16S Barcode, and 14 μl of Nuclease-free water. Thermal cycling comprised an initial denaturation step (1 min at 95 ˚C), followed by 25 cycles of denaturation (20 sec at 95 ˚C), annealing (30 sec at 55˚C), and 2 min extension at 65 ˚C. The final extension was performed for 5 min at 65 ˚C.”, “(FLO-MIN106D, Oxford Nanopore Technology, United Kingdom)”, “workflow v2022.01.07”, and “(Oxford Nanopore Technologies, United Kingdom) with default settings” in Materials and methods of metagenomic analysis and highlighted in yellow in the first paragraph on page 9. No statistical analysis was used in this study.
We think the final decision on whether to categorize this manuscript as a short report or an original article is up to the Editor-in-Chief. We are confident that the detailed verification results using novel molecular biological technique with comparing with conventional examinations of the 8 cases are suitable for an original article.
Reviewer 2 Report
This paper examines and compares the usefulness of scraping, culture and metagenomics for identification of bacteria causing keratitis. The authors conclude that scraping is still important and can provide fairly adequate data on which to base antimicrobial therapeutic choices. The benefit of metagenomics was seen with only one case (care 7), and perhaps case 6 (if genera rather than species is taken into consideration?). This is of interest in the field, and supports current clinical practice.
Some changes are recommended below:
change “prognosis differ depend on it.” to “prognosis differ depending on the causative agent.”
change “from specimen, indicating its survival” to “from specimens, indicating their survival”
change “Therefore, a specific bacterium is isolated from the corneal scraping or eye dis- 44
charge, a diagnosing bacterial keratoconjunctivitis caused by it.” to “When a specific bacterium is isolated from the corneal scraping or eye charge, a specific diagnosis of bacterial keratoconjunctivitis can be made.”
change “susceptibility was also determined” to “susceptibility is also determined”
change “selection of antimicrobial agent.” to “selection of antimicrobial agents.”
change “large amount of the causative bacterium from the specimen” to “large amounts of material”
Table 1 - change “P. aeruginsa” to “P. aeruginosa”; change “Str. pneumonia” to “Str. pneumoniae”
Figure 1C - the genera and species have been constricted together so they read as one name - amend to add space between genera and species
“Case 7 also produced results highlighting contamination, which is a weakness of the 217
metagenomic analysis.” - I’m not sure on what basis the statement about “contamination” is made - although reading further down in the discussion there is some mention of N. gonorrhoeae. There is a clear abundance of S. aureus, and so even if there was a suspicion of contamination, not proven, a positive diagnosis could still be made based on metagenomics. I think the very small % of Neisseria and possible contamination is over stated in this case.
“results of Case 6 appeared to be contaminated” Cases of keratitis can be polymicrobial - although admittedly that is rare - but couldn’t this be one such case? The case for possible contamination is better made for Case 6.
Author Response
2) Comments and Suggestions for Authors
This paper examines and compares the usefulness of scraping, culture and metagenomics for identification of bacteria causing keratitis. The authors conclude that scraping is still important and can provide fairly adequate data on which to base antimicrobial therapeutic choices. The benefit of metagenomics was seen with only one case (care 7), and perhaps case 6 (if genera rather than species is taken into consideration?). This is of interest in the field, and supports current clinical practice.
Some changes are recommended below:
change “prognosis differ depend on it.” to “prognosis differ depending on the causative agent.”
Response: We have revised it, as suggested.
change “from specimen, indicating its survival” to “from specimens, indicating their survival”
Response: We have revised it, as suggested.
change “Therefore, a specific bacterium is isolated from the corneal scraping or eye discharge, a diagnosing bacterial keratoconjunctivitis caused by it.” to “When a specific bacterium is isolated from the corneal scraping or eye charge, a specific diagnosis of bacterial keratoconjunctivitis can be made.”
Response: We have revised it, as suggested.
change “susceptibility was also determined” to “susceptibility is also determined”
Response: We have revised it, as suggested.
change “selection of antimicrobial agent.” to “selection of antimicrobial agents.”
Response: We have revised it, as suggested.
change “large amount of the causative bacterium from the specimen” to “large amounts of material”
Response: We have revised it, as suggested.
Table 1 - change “P. aeruginsa” to “P. aeruginosa”; change “Str. pneumonia” to “Str. pneumoniae”
Response: We have addressed all the spelling errors
.
Figure 1C - the genera and species have been constricted together so they read as one name - amend to add space between genera and species
Response:
The Word file we submitted included an Excel generated pie chart and legend. In the original Excel legend, there is a one-byte space between species and genus which may be difficult to distinguish because the characters are small. We have uploaded the same PDF and the Word file to the submission web site for your review.
“Case 7 also produced results highlighting contamination, which is a weakness of the metagenomic analysis.” - I’m not sure on what basis the statement about “contamination” is made - although reading further down in the discussion there is some mention of N. gonorrhoeae. There is a clear abundance of S. aureus, and so even if there was a suspicion of contamination, not proven, a positive diagnosis could still be made based on metagenomics. I think the very small % of Neisseria and possible contamination is over stated in this case.
Response:
As you pointed out, we think Case 7 was identified as S. aureus keratitis, which was diagnosed using metagenomic analysis. Therefore, at the beginning of the third paragraph of the "Discussion" section, we mentioned the usefulness of metagenomic analysis in this case. Due to its pathogenicity, it is well-known fact that even in a small amount of the bacteria Neisseria gonorrhoeae induces severe keratoconjunctivitis. However, based on the clinical findings and course, Case 7 is not a gonococcal infection. From the above, Case 7 was completely ruled out of gonococcal infection, and Case 8 was conclusively gonococcal infection. Samples of these two cases were each collected at quite different times; however processed on the same day. Therefore, we think it is most likely contamination during the sample manipulation. Thus, we suggest that this contamination case requires consideration when interpreting the results of sensitive metagenomic analysis of ocular sample which has few microbial loads.
“results of Case 6 appeared to be contaminated” Cases of keratitis can be polymicrobial - although admittedly that is rare - but couldn’t this be one such case? The case for possible contamination is better made for Case 6.
Response:
The possibility of a polymicrobial infection certainly cannot be completely denied in Case 6. However, as we have described in the fourth paragraph of "Discussion" section, S. alvi has only been reported as an intestinal bacterium of honeybees, and to the best of our knowledge, no scientific report of S. alvi eye infections has been recorded. Most bacteria other than S. alvi detected using metagenomic analysis were typical ocular surface and/or eyelids flora, such as Corynebcterium and Moraxella. Although Corynebacterium was isolated from the culture, it is unlikely to that these gram-positive bacteria caused keratitis in Case 6, as only a small quantity of gram-positive rods was seen in smear microscopy,
If the concern that "The case for possible contamination is better made for Case 6" is related to the findings that one read (1.1%) of Neisseria gonorrhoeae DNA was also detected in Case 6, we would like to explain this further. Case 6 was processed on a different day than Cases 7 and 8; however, they were processed within10 days of each other. Sample contamination is possible when working with the same equipment of a laboratory over time. However, as S. alvi was the most abundantly detected bacteria in Case 6, we omitted any detailed descriptions to avoid any confusion. In the revised manuscript, the last part of the 4th paragraph of the "Discussion" was revised as follows:
“The specimens from Cases 6 and 7 contained only one read (relative abundance of 1.1% and 1.6%, respectively) of Neisseria gonorrhoeae DNA. Although the sample collection dates for Cases 6, 7, and 8 were different, the same examiner (HE) performed the DNA extraction from the specimens and library preparation for the MinION for 10 days (on the same day for Cases 7 and 8). It is presumed that specimens from Cases 6 and 7 were contaminated with the gonococcal DNA of the Case 8 sample during experimental manipulation.”
Round 2
Reviewer 1 Report
No further comments.